# Risk-taking incentives predict aggression heuristics in female gorillas

**Nikolaos Smit[1,2]\*, Martha M Robbins[2]**

[1]Department of Biology, University of Turku, Turku, Finland; [2]Max Planck Institute for Evolutionary Anthropology, Leipzig, Germany

## eLife Assessment

This **important** study uses long-term behavioural observations to understand the factors that influence female-on-female aggression in gorilla social groups. The evidence supporting the claims is **convincing**, as it includes novel methods of assessing aggression and considers other potential factors. The work will be of interest to broad biologists working on the social interactions of animals.

## Abstract

Competition is commonly reflected in aggressive interactions among groupmates as individuals try to attain or maintain higher social ranks that can offer them better access to critical resources. In this study, we investigate the factors that can shift competitive incentives against higher- or lower-ranking groupmates, that is, more or less powerful individuals. We use a long-term behavioural data set on five wild groups of the two gorilla species starting in 1998, and we show that most aggression is directed from higher- to lower-ranking adult females close in rank, highlighting rank-reinforcement incentives. Yet, females directed 42% of aggression to higher-ranking females than themselves. Females targeted groupmates of higher rank with increasing number of males in the group, suggesting that males might buffer female–female aggression risk. Contrarily, they targeted females of lower rank with increasing number of females in the group, potentially because this is a low-risk option that females prefer when they have access to a larger pool of competitors to choose from. Lactating and pregnant females, especially those in the latest stage of pregnancy, targeted groupmates of higher rank than the groupmates that cycling females targeted, suggesting that energetic needs may motivate females to risk confrontation with more powerful rivals. Our study provides critical insights into the evolution of competitive behaviour, showing that aggression heuristics, the simple rules that animals use to guide their aggressive interactions, are not merely species-specific but also dependent on the conditions that populations and individuals experience.

**\*For correspondence:**
nismit@utu.fi

**Competing interest:** The authors declare that no competing interests exist.

## Introduction

Animals that live in groups often compete for access to resources such as food and mates (*Carel, 1989*; *Koenig, 2002*). The potential costs of this competition can drive the formation of hierarchies that determine priority of access to resources without superfluous conflicts (*Rowell, 1974*; *Boone, 2017*). Accordingly, individuals may choose strategically who they compete with in order to minimize costs and maximize gains. Previous research suggests that individuals primarily compete with those closest in the hierarchy, to attain (against close higher-ranking) or maintain (against close lower-ranking individuals) their ranks ('rank-dependent aggression'; *Vogel et al., 2007*; *Hobson and DeDeo, 2015*; *Wright et al., 2019*; *Hobson et al., 2021*; *Dehnen et al., 2022b*; *Dehnen et al., 2022a*). Aggression, probably the most straightforward proxy of competition, commonly increases in frequency when resource availability is lower, and it is most often directed towards lower-ranking individuals (*Hobson et al., 2021*; *Smit and Robbins, 2024*). Yet, recent research has demonstrated that the rules that

**eLife digest** In the wild, animals frequently compete for access to resources critical for survival and reproduction, such as food. Competition among members of a social group often leads to social hierarchies, with higher-ranking individuals typically acting as aggressors, and conflicts occurring mostly between individuals of similar rank. While patterns of aggression are often considered species-specific, social, ecological, and physiological factors may also play important roles in shaping them.

To better understand the dynamics underlying aggressive interactions in animals, Smit and Robbins analysed long-term data sets of five groups of wild gorillas, dating back to the late 1990s. These included one group of wild western gorillas and four groups of wild mountain gorillas. They recorded aggressive interactions among females and assigned each a 'risk score'. Interactions were considered riskier when lower-ranking females targeted higher-ranking opponents.

The analyses revealed that females with greater energetic needs – such as pregnant and lactating females – were more likely to engage in risky aggression, targeting more powerful opponents. Females also directed aggression toward more powerful opponents when more males were present, which could protect them from retaliation. Conversely, females chose weaker opponents when more females were present, suggesting that when females have more options, they choose among the less risky ones.

These results suggest that animals can adapt their behaviour based on both their social environment and individual needs. As a result, aggression tactics and social dynamics can vary significantly even within the same species – driven by the circumstances that individuals experience at different times.

Studying competition in animal societies and aggression among individuals of different ranks offers valuable insights into the evolution of more egalitarian or despotic structures – including in humans. This research can also help us better understand the behaviour of individuals experiencing resource scarcity or desperation, and how these can motivate individuals to navigate their social landscape and challenge their social hierarchies.

animals use to guide their aggressive interactions ('aggression heuristics') towards groupmates of different ranks vary even within species (*Hobson and DeDeo, 2015*; *Hobson et al., 2021*). In this study, we test the hypothesis that this variation arises due to the different conditions that (individual) animals experience, and specifically that the social environment and individual energetic needs shape competitive incentives and aggression towards individuals of different ranks.

Regarding the social environment, larger group sizes may entail a larger number of low-ranking individuals which are cumulatively targeted, as competition and aggression are preferentially directed towards the lowest-ranking groupmates (see also 'bullying' in *Hobson et al., 2021*). When individuals have access to a larger pool of competitors, they may preferentially target less powerful ones, consistent with risk-sensitivity theory, which posits that individuals tend to minimize risks when they have the option (*Caraco, 1981*; *Barclay et al., 2018*). Conversely, larger group sizes might entail a larger number of allies (e.g. kin for spotted hyaenas, *Crocuta crocuta*, *Vullioud et al., 2019*) or protectors (e.g. adult males for female gorillas, *Watts, 1997*; *Smit and Robbins, 2024*), which can increase social support and, eventually, minimize any risk-related costs of aggression towards higher-ranking competitors. Hence, group size, the overall number of individuals in the group, might be a poor predictor of aggression because it conflates opposing effects of different kinds of individuals (e.g. see *Smit and Robbins, 2024* for an example of opposing effects of the number of females and number of males on female gorilla aggression). Finally, if larger group sizes promote within-group competition increasing overall aggression rates (*Slotow, 1996*; *Koenig, 2002*; *Smit and Robbins, 2024*), they might simultaneously promote high-ranking individuals to reinforce their status by targeting lower-ranking groupmates and lower-ranking individuals to direct aggression against higher-ranking groupmates if this can allow them access to precious resources.

Regarding the energetic needs, greater needs for resources may boost the incentives of higher-ranking individuals to reinforce their status by strategically directing their competitive efforts towards lower-ranking groupmates (*Hobson et al., 2021*; *Dehnen et al., 2022b*; *Strauss and Shizuka, 2022*). Contrarily, greater individual needs may also prompt low-ranking individuals, who struggle more to

access resources and experience a lower risk-reward ratio (more to gain, less to lose), to show greater aggression rates even against higher-ranking groupmates (*Parker and Rubenstein, 1981*; *Sapolsky, 1993*; *Sapolsky, 2005*). This aggression may help individuals to improve their ranks but it might be risky if it can incite retaliation from high-ranking, powerful, recipients. Yet, the benefits related to status improvement (long-term benefit) or resource acquisition (short-term benefit) might counterbalance any risk-related costs. Various empirical examples support the latter hypothesis: hunger-driven payoff asymmetries can increase aggression from lower- to higher-ranking individuals (noble crayfish, *Astacus astacus*; *Gruber et al., 2016*), reproductive suppression of low-ranking females can promote conflict escalation against high-ranking ones (paper wasps, *Polistes dominulus*; *Cant et al., 2006*), and the energetic/nutritional needs of pregnancy (due to support of fetal growth) or lactation (due to milk production) can increase female aggression that reverses hierarchical relationships (*Murie and Harris, 1988*; *Lu et al., 2013*; *Lu et al., 2016*).

Studies investigating the influence of social, ecological, or physiological factors on aggression patterns across species usually focus on aggression frequency/rate (*Vogel and Janson, 2011*; *Nie et al., 2013*); in this study, we build on this literature to test how relevant factors influence 'aggression direction' in terms of power differentials, aiming to unravel another evolutionary aspect of competitive strategies. Gorillas represent an intriguing case for this endeavour because females of both species form surprisingly stable hierarchical relationships, usually maintained over their whole co-residence in a group, but they often direct aggression towards higher-ranking groupmates (*Smit and Robbins, 2025*; *Watts, 1994*; *Robbins, 2008*). Female–female aggression rates towards both higher- and lower-ranking rivals decrease with the number of adult males/protectors in the group but increase with the number of females/competitors in the group (*Watts, 1994*; *Robbins, 2008*; *Wright and Robbins, 2014*; *Smit and Robbins, 2024*). Additionally, aggression rates are greater in pregnant females (*Smit and Robbins, 2024*), which, like lactating females, spend more time feeding (*Watts, 1983*), highlighting the greater energetic needs of females in these reproductive states – similar to humans and other apes (*Dufour and Sauther, 2002*).

We use behavioural observations on one wild western (*Gorilla gorilla gorilla*) and four wild mountain (*Gorilla beringei beringei*) gorilla groups, starting in 1998 in one of the mountain gorilla groups, to test if the social environment or energetic needs influence female aggression towards more or less powerful females. Specifically, we test whether females direct aggression of lower or higher score (score = recipient-aggressor rank difference) depending on (i) the number of males in their group who may support or protect females, (ii) the number of females in the group representing competitors over resources, and (iii) female reproductive state (cycling, pregnant, or lactating) which is an indirect proxy of energetic needs. We compare our results on the direction of aggression to previous results examining the effects of the same variables on aggression frequency/rates.

## Methods
### Study system and behavioural data
We studied one western gorilla group (ATA/Atananga; *Table 1*) in Loango National Park, Gabon, and four mountain gorilla groups in Bwindi Impenetrable National Park, Uganda (*Table 1*). Observations of western gorillas lasted typically between 07:00 and 16:30 h but observations of mountain gorillas were limited to 4 hours per day, typically between 08:00 and 15:00 h following the regulations of the Uganda Wildlife Authority.

**Table 1.** Study groups, study period, average number of females per day (± s.d.; total number of females in the group in parentheses), and number of aggressive interactions (mild - moderate - severe).

| Group name | Study period | Number of females | Aggression (mild-moderate-severe) |
| --- | --- | --- | --- |
| ATA/Atananga | 2017–2023 | 3.30±0.85 (6) | 208 - 65 - 27 |
| BIT/Bitukura | 2015–2023 | 4.14±0.35 (6) | 509 - 35 - 23 |
| KYA/Kyagurilo | 1999–2016 | 6.07±1.02 (9) | 3256 - 445 - 315 |
| MUK/Mukiza | 2016–2023 | 6.59±1.20 (8) | 1389 - 89 - 141 |
| ORU/Oruzogo | 2014–2023 | 5.59±0.96 (8) | 55 - 7 - 7 |

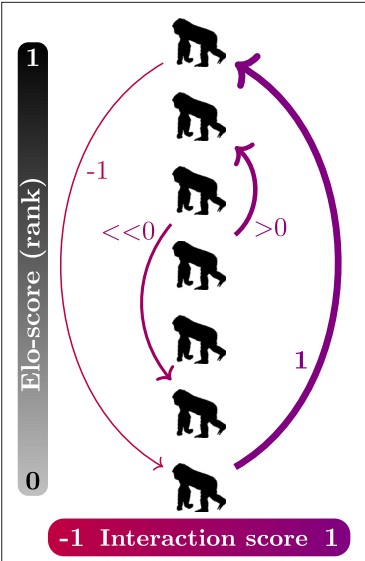

**Figure 1.** Calculation of the interaction score. The lower the rank of the aggressor and the greater the rank of the recipient, the greater the score (-1 to 1; line thickness). Arrows start from aggressors and point to recipients of aggression. We created the figure using a female gorilla silhouette icon from http://phylopic.org (created by T. Michael Keesey) and TikZ (TeX).

Trained observers recorded both focal and ad libitum behavioural observations, including decided avoidance (when an individual walks away from another approaching individual) and displacement (when an individual avoids another and the latter takes the place of the former) behaviours. These two behaviours are ritualized, occurring in the absence of aggression, they are considered a more reliable proxy of power relationships over aggression (*Smit, 2024*), and they are typically used to infer gorilla hierarchical relationships (*Watts, 1994*; *Robbins, 2008*). Similar to the recent studies which have inferred social ranks in gorillas (*Wright et al., 2020*; *Smit and Robbins, 2024*), we used all avoidance and displacement interactions throughout the study period and the function *elo.seq* from R package *EloRating* (*Neumann et al., 2011*) to infer daily individual female Elo-scores (*Elo, 1978*; *Albers and de Vries, 2001*). We assigned to all avoidance/displacement interactions equal intensity, that is, equal influence to the power relationship of the interacting individuals (k=100). We also assigned to individuals present at the onset of the study initial Elo-scores of 1000 and to individuals entering the hierarchy later (e.g. maturing individuals or immigrants) the score of the lowest ranking individual during the entrance day (*Smit and Robbins, 2024*). This method takes into account the temporal sequence of interactions and updates an individual's Elo-scores each day the individual interacted with another. The Elo-scores of the winner and loser of an interaction are updated as a function of the winning probabilities prior to the interaction: winners with low winning probabilities get greater score increases than winners with high winning probabilities and losers with low winning probabilities get smaller score decreases than losers with high winning probabilities. We present these interactions and hierarchies in detail in *Smit and Robbins, 2025* (see 'traditional Elo rating method'; we do not use the 'optimized Elo-rating method' as it yields similar hierarchies and it is not widely used). We standardized Elo-scores per group and day such that the highest score was 1 and the lowest 0.

The observers also recorded aggressive behaviours among adult females (>10 years old for western; >8 years old for mountain female gorillas; *Smit and Robbins, 2024*), which can be context dependent (*Smit, 2024*). For our analysis, we classified these behaviours into three intensity categories (as per *Robbins, 1996*; *Robbins, 2008*): mild (cough/pig grunt [aggressive vocalization that sounds like a pig grunting or a deep throated cough], soft bark [characteristic aggressive vocalization], bark, scream, and pull vegetation), moderate (chest-beat [beating of chest by hands], strut-stand/run [stand in front of or runs past another with a distinctive stiff leg stance], lunge [aggressively and quickly leaning forward towards another], direct charge, indirect charge, run at/over, push) and severe aggression (hit [aggressively strike/slap/push, etc., another], attack [jumps on, bites, etc., another], drag [pulls another along the ground], fight [two individuals attack, bite, scream, etc., at each other], bite, chase, kick). To quantify 'aggression direction', we assigned a score to each aggressive interaction, calculated by subtracting the standardized Elo-score of the aggressor from that of the recipient. The score was maximum (=1) for interactions where the aggressor was the lowest-ranking female and the recipient was the highest-ranking female; and minimum (=-1) for interactions where the aggressor was the highest-ranking female and the recipient was the lowest-ranking female. More generally, positive scores represented aggression up and negative scores represented aggression down the hierarchy (*Figure 1*).

We used demographic data to estimate daily female reproductive state. On a given day, we classified as 'pregnant' any female that gave birth 255 days or less after that day (*Czekala and Sicotte, 2000*), as 'cycling' any female that was not classified as pregnant and she had been observed mating since her last parturition and as 'lactating' any female with a dependent infant (based on last observation of nipple contact; *Eckardt et al., 2016*; *Robbins and Robbins, 2021*) that was not pregnant and had not observed mating since her last parturition. Lactation is often considered more energetically demanding than pregnancy as a whole but the latest stages of pregnancy are highly energetically demanding, potentially even more than lactation (*Butte and King, 2005*; *Noren et al., 2014*). Thus, we differentiated between the first (1–85th day of pregnancy), second (85–170th day), and third (170–255th day) trimester of pregnancy (85 days each).

## Statistical analyses

We fitted a linear mixed-effects model with a logit function to test whether females who have greater energetic needs and/or experience different social environments, direct aggression of higher or lower score (response variable, continuous, between –1 and 1; *Figure 1*) to other females. Given that we tested for the aggression direction and not aggression rates, the design of our analysis was independent of observation effort, and thus, we were able to use both focal and ad libitum observations. Specifically, we considered each aggressive interaction recorded during either a focal or an ad libitum observation as a separate data point. In our model, we fitted the following explanatory variables: aggression intensity to test if aggression of greater score is more often mild than moderate or severe; number of adult males in the group (>14 years old for western males; >12 years old for mountain males); number of females in the group; reproductive state of the aggressor (cycling, trimester of pregnancy, or lactation); and species (western or mountain). We fitted the identities of interacting females, dyad, and group as random factors. Finally, we used Tukey post hoc comparisons (via the *glht* function from the *multcomp* package; *Hothorn et al., 2025*), to perform pairwise comparisons between all reproductive states.

We ran the model in R version 4.1.2 using the function *glmmTMB* from the package *glmmTMB* (*Brooks et al., 2017*). We used the function *Anova* from package *car* (*Fox and Weisberg, 2019*) to test the significance of fixed factors and to compute 95% confidence intervals. We tested the residual distributions using the functions *testDispersion* and *testUniformity* from package *DHARMa* version 0.4.6 (*Hartig, 2022*) to validate the model. We used the base function *cor.test* and the function

**Table 2.** Results from the linear mixed-effects model.

Significant p-values appear in bold. The significance of each level of a categorical variable was evaluated against the reference level (placed in parenthesis) according to whether their 95% confidence intervals (CI) include zero or not. 'Pregnant_n' denotes the nth trimester of pregnancy. To highlight that aggression rates can increase due to an increase in interactions of different score, we include a last column with the effect of some of the tested variables on overall adult female aggression rates, based on results of linear mixed effects models from *Smit and Robbins, 2024*. 'ns': non-significant correlation; '+': positive correlation; '-': negative correlation; 'na': not tested (see *Smit and Robbins, 2024* for details).

**Response variable: Interaction score (recipient-aggressor rank difference)**

| Fixed factor | Level | Estimate | 95% CI | Chisq | p-Value | Aggression Rate (from Smit and Robbins, 2024) |
|---|---|---|---|---|---|---|
| Species (mountain) | Western | 0.088 | [–0.495; 0.671] | 0.088 | 0.767 | ns |
| | Pregnant_1 | **0.130** | [0.101; 0.159] | 138.812 | **<0.001** | +* |
| | Pregnant_2 | **0.093** | [0.063; 0.123] | 138.812 | **<0.001** | +* |
| | Pregnant_3 | **0.146** | [0.114; 0.177] | 138.812 | **<0.001** | +* |
| Reproductive state (Cycling) | Lactating | **0.043** | [0.027; 0.059] | 138.812 | **<0.001** | - |
| Number of females | | **–0.011** | [–0.018; –0.003] | 8.045 | **0.005** | + |
| Number of males | | **0.018** | [0.010; 0.026] | 19.784 | **<0.001** | - |
| | Moderate | **–0.030** | [–0.051; –0.009] | 8.531 | **0.005** | na |
| Aggression intensity (Mild) | Severe | –0.014 | [–0.036; 0.009] | 8.531 | 0.229 | na |

*Pregnancy was not divided in trimesters in *Smit and Robbins, 2024*.

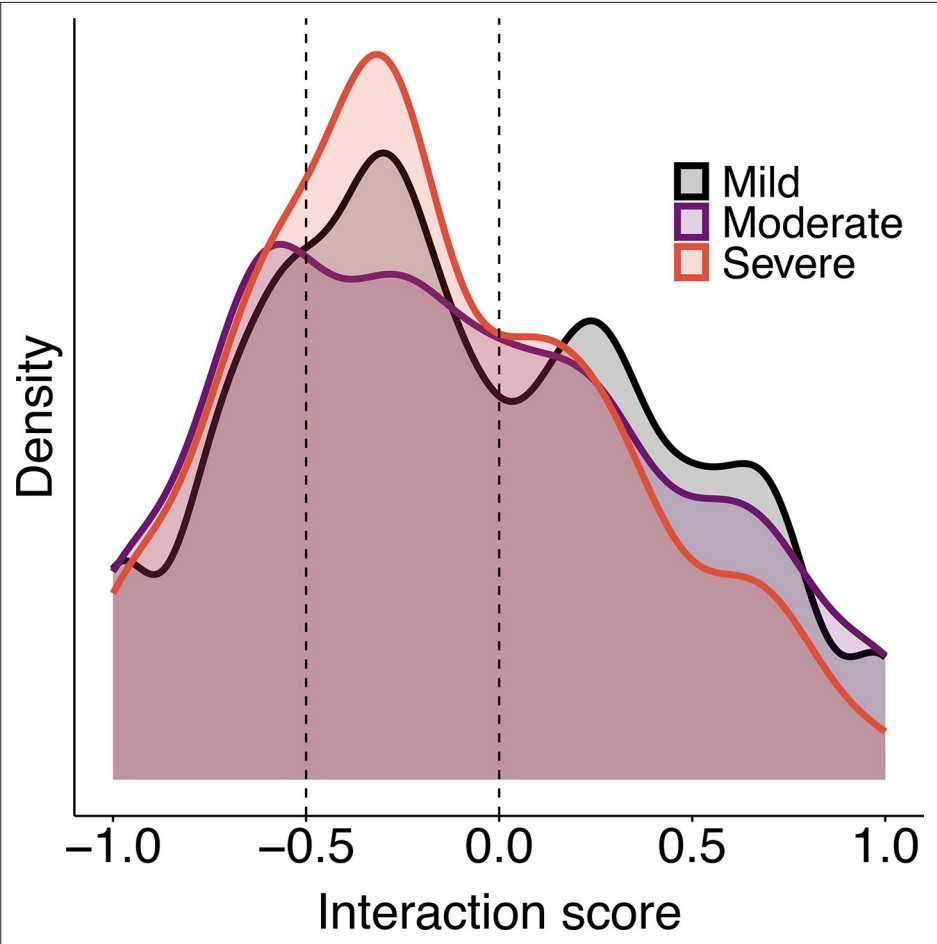

**Figure 2.** Distribution of interaction score (recipient-aggressor rank difference). Density of the mild, moderate, and severe aggression as a function of the interaction score. Positive scores represent aggression up and negative scores represent aggression down the hierarchy.

*check_collinearity* from the *performance* package to test for correlations and multicollinearities of the explanatory variables: all variance inflation factor values were <1.5 indicating no serious multicollinearities (*Zuur et al., 2010*).

## Results

We analysed 6871 aggressive interactions among a total of 31 adult female gorillas in the five social groups (*Table 2*). The average percentage of aggressive interactions directed from lower- to higher-ranking females across groups was 41.8 ± 6.7% (± SD; ATA: 47.2%; BIT: 46.3%; KYA: 36.5%; MUK: 46.1%; ORU: 32.7%). For comparison, only 16.4 ± 4.1% of displacement/avoidance interactions that we used to infer the highly stable hierarchies (details in *Smit and Robbins, 2025*) were directed from lower- to higher-ranking females (± SD; ATA: 15.4%; BIT: 19.2%; KYA: 12.8%; MUK: 22.1%; ORU: 12.5%). This result confirms previous evidence that aggression may not be a reliable proxy of power in gorillas or other species (*Watts, 1994*; *Robbins, 2008*; *Smit, 2024*).

Aggression from lower- to higher-ranking females was 85% mild, 9% moderate, and 6% severe while aggression from higher- to lower-ranking females was 82% mild, 10% moderate, and 8% severe. Generally, aggression of different intensity showed similar distributions and all aggression was most common from higher- to lower-ranking females close in rank (–0.5>score>0; *Figure 2*). The interactions of mild aggression were of greater score (recipient-aggressor rank difference) than the interactions of moderate aggression, meaning that females were more likely to use mild rather than moderate

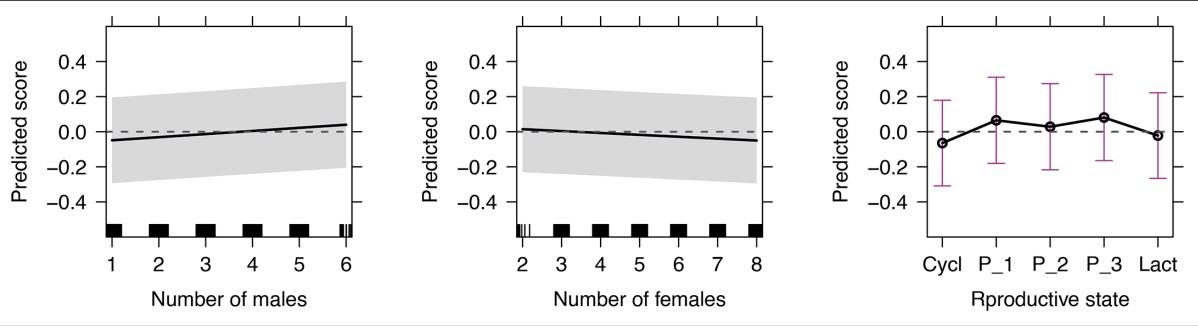

**Figure 3.** Predicted interaction score (recipient-aggressor rank difference) of aggressive interactions (n=6871) as a function of the explanatory variables of the linear mixed effects model with a significant effect: number of adult males in the group, number of adult females in the group and aggressor's reproductive state (Cycl: cycling; P_n: nth pregnancy trimester; Lact: lactating). Shaded areas and whiskers show 95% confidence intervals. We created the figure using R package *effects* (**Fox et al., 2022**). Positive scores represented aggression up and negative scores represented aggression down the hierarchy.

aggression against more powerful rivals, but the score difference between interactions of severe and moderate or mild aggression was not significant (**Table 2**).

Female gorillas directed aggression of greater score when there were more males in the group and when there were fewer females in the group (**Figure 3**, **Table 2**). When we ran our analysis testing for group size (number of weaned individuals in the group), instead of the numbers of males and females, its influence on interaction score was not significant (estimate=−0.001, p-value=0.682). Females in the third trimester of pregnancy directed the aggression of the greatest score, females in any pregnancy stage directed aggression of greater score than lactating and cycling females, and lactating females directed aggression of greater score than cycling females, likely highlighting an effect of energetic needs on aggression heuristics (**Figure 3**, **Table 2**). Post hoc pairwise comparisons showed significant differences between all reproductive states, apart from the difference of the first and the other two trimesters of pregnancy (**Table 3**, **Figure 3**). When we ran our analysis merging pregnancy trimesters into one category, we found pregnant females to direct aggression of significantly greater score than lactating and cycling females, and lactating females to direct aggression of significantly greater score than cycling females (not shown). Finally, our results did not show any significant difference between aggression score of western (only one group) and mountain gorillas groups.

Notably, a positive (or negative) correlation of a predictor with the interaction score does not necessarily represent a shift from aggression towards females lower-ranking than the aggressor to aggression towards females higher-ranking than the aggressor (or vice versa) – but, more generally,

**Table 3.** Results of post hoc pairwise comparisons among reproductive states, from the linear mixed effects model.

Significant p-values appear in bold. Cycl: cycling; P_n: nth pregnancy trimester; Lact: lactating.

| Comparison | Estimate | Std. error | Z-value | p-Value |
|---|---|---|---|---|
| P1-Cycl | 0.130 | 0.015 | 8.681 | **0.000** |
| P_2 - Cycl | 0.093 | 0.015 | 6.071 | **0.000** |
| P_3 - Cycl | 0.146 | 0.016 | 9.147 | **0.000** |
| Lact - Cycl | 0.043 | 0.008 | 5.236 | **0.000** |
| P_1 - P_2 | −0.037 | 0.019 | −1.965 | 0.267 |
| P_1 - P_3 | 0.016 | 0.019 | 0.812 | 0.922 |
| Lact - P_1 | −0.087 | 0.015 | −5.888 | **0.000** |
| P_2 - P_3 | 0.052 | 0.019 | 2.721 | **0.047** |
| Lact - P_2 | −0.050 | 0.015 | −3.368 | **0.006** |
| Lact - P_3 | −0.103 | 0.015 | −6.715 | **0.000** |

it represents a shift of aggression towards more (or less) powerful females independently of the rank relationship to the aggressor. For example, females in the second trimester of pregnancy direct most aggression towards rivals higher-ranking than themselves; yet they direct aggression of significantly lower score than females in the third trimester of pregnancy, that is, the latter direct more aggression to females even more higher-ranking than those during the second trimester (*Figure 3*, *Table 2*).

## Discussion

Most aggression was directed from higher to lower-ranking females, usually close in rank, supporting the hypothesis that individuals commonly use aggression to reinforce their status. However, approximately 42% of aggression was directed from lower- to higher-ranking females, which is even greater than previous estimations in gorillas (34% [*Watts, 1994*] and 25% [*Robbins, 2008*]) and also greater than in many other animals (*Hobson et al., 2021*). Our results suggest that such aggression towards more powerful rivals reflects competitive incentives influenced by the social environment and driven by circumstantial needs. This interpretation is in line with previous observations showing that female gorillas are usually unable to improve their ranks through active competition, as they form highly stable hierarchical relationships (*Smit and Robbins, 2025*), that is, aggression is unlikely to be used for challenging the hierarchy. Importantly, female gorillas are more likely to respond aggressively ('retaliate') than submissively to aggression from other females (*Watts, 1994*), meaning that aggression involves some risk for the aggressor, especially when it targets more powerful groupmates. Thus, greater interaction scores reflect greater risks (the greater the relative rank of the recipient, the greater the risk for the aggressor), and our results suggest that the social environment and circumstantial needs may influence individual decisions to engage in risky behaviours.

Female gorillas appeared to direct aggression of greater score when there were more males in the group, potentially supporting our interpretation that male support or protection (*Watts, 1994*; *Watts, 1997*) provides females with an environment to take greater risks. Males may support lower-ranking females in order to decrease competitive inequities among females (e.g. to prevent low-ranking female emigration from their group; *Sicotte, 2002*) and high-ranking females might hesitate to retaliate to aggressors and escalate a contest if more males are adjacent to the aggressors or are simply present in the group and can intervene. Interestingly, when females have access to fewer males, they generally exhibit greater aggression rates towards other females (*Smit and Robbins, 2024*; *Table 2*, column 'Aggression rates'), potentially competing for male protection per se; but once they have access to more males (and male protection), they appear to direct more aggression to more powerful female rivals.

In contrast to the number of adult males, the number of adult females in the group was negatively correlated with interaction score, that is, female gorillas directed aggression to lower-ranking, less powerful, females when there were more females in the group. Hence, the previously observed increase in aggression rates in groups with more females (*Smit and Robbins, 2024*; *Table 2*, column 'Aggression rates') likely pertains predominantly to aggression from more to less powerful rivals. Our result may reflect that females preferably target less powerful rivals when they have access to a larger pool of competitors to choose from, similar to humans (*Savage et al., 2020*; see also risk-sensitivity theory in introduction). Alternatively, the increased competition due to a larger number of competitors may prompt higher-ranking females to reinforce their status more than it prompts lower-ranking females to target higher-ranking ones in order to access resources. Overall, the combination of our present and previous results (*Smit and Robbins, 2024*) showing the influence of the number of males and females on both female aggression rates and aggression direction confirms that non-human animals can adapt their aggression patterns according to the social context or the available social information (see also *Haux et al., 2021*).

Aggression score was also influenced by energetic needs: female gorillas in the most energetically demanding reproductive states directed aggression to more powerful females (greater aggression score = greaterrecipient-aggressor rank difference). Females in the last and most energetically demanding stage of pregnancy directed aggression of greater score than all other females (this difference was significant for all reproductive states except for females in the first trimester of pregnancy), females in any stage of pregnancy directed aggression of greater score than lactating and cycling females, and lactating females directed aggression of greater score than cycling females. While lactating females can have greater energetic needs than pregnant females, they might direct

aggression of lower score than pregnant females because they have lower risk tolerance in aggression, especially towards higher-ranking groupmates, in order to protect their dependent infants, reminiscent of risk-avoidance behaviours of lactating female African wild dogs (*Lycaon pictus*; *Marneweck et al., 2021*) and black bears (*Ursus americanus*; *Gantchoff et al., 2019*). This interpretation is consistent with previous results showing that lactating females show the lowest aggression rates (*Smit and Robbins, 2024*; see also column 'Aggression rates' in *Table 2*). Yet, lactating females directed aggression of greater score than cycling females, despite the fact that they exhibit lower aggression rates than cycling females (*Smit and Robbins, 2024*; *Table 2*, column 'Aggression rates'). Lactating females have greater proximity to alpha males (*Harcourt, 1979*; *Rosenbaum et al., 2016*), that potentially offers them greater access to resources with no need for direct female-female competition (low aggression rates), but it might also buffer female–female aggression risk if males mediate/intervene in female aggressive interactions (*Watts, 1994*; *Watts, 1997*), allowing lactating females to direct aggression to generally more powerful rivals (high aggression score), even if this aggression is infrequent.

Our study suggests that aggression heuristics depend on the social environment and individual needs, meaning that the variation in heuristics observed within or between species (*Hobson et al., 2021*) at least partially reflects differences in the conditions that specific animal populations or individuals experience at different time points. Our study also adds to behavioural observations from several species suggesting that individuals with greater needs might engage in more risky behaviours (*Verdolin, 2006*; *Bruyneel et al., 2009*; *Jordan et al., 2011*), including inter-individual competition (*Garamszegi et al., 2009*; *Rosati and Hare, 2012*; *Bertram et al., 2016*). Accordingly, it improves our understanding of the evolution of risk-taking in hominids, including (aggressive) competition in humans where individuals unsuccessful in economic or mating competition may exhibit risky aggressive behaviours (*Wilson and Daly, 1985*; *Campbell, 1995*; *Mishra and Lalumière, 2008*; *Hill and Buss, 2010*; *Wohl et al., 2014*). Finally, our results may provide some insights regarding the evolution of more egalitarian or despotic societies of other species: if certain social factors or individual conditions can drive shifts in aggression up or down the hierarchy, then they have the potential to flatten or reinforce the hierarchy, respectively.

## Acknowledgements

We thank all staff who assisted with data collection, project management, and logistical support in both study sites (see https://www.eva.mpg.de/primate-behavior-and-evolution/research-groups/gorilla-group/). We also thank Andrew M Robbins and Fernando Colchero for the useful feedback in the course of this project, Jack L Richardson and Christopher Young for long-term database management and three anonymous reviewers and the editorial team for constructive comments. We thank the Uganda Wildlife Authority and the Ugandan National Council for Science and Technology for permission to work in Bwindi Impenetrable National Park, Uganda, as well as the Institute of Tropical Forest Conservation for logistical support in Bwindi. We thank the Agence Nationale des Parcs Nationaux and the Centre National de la Recherche Scientifique et Technique of Gabon for permission to work in Loango and for the help in project management.

## Additional information

### Funding

| Funder | Grant reference number | Author |
|---|---|---|
| Max Planck Society | | Martha M Robbins |
| United States Fish and Wildlife Service | | Martha M Robbins |
| Great Ape Fund | | Martha M Robbins |
| Tusk Trust | | Martha M Robbins |
| Taipei Zoo | | Martha M Robbins |

| Funder | Grant reference number | Author |
| --- | --- | --- |
| Berggorilla & Regenwald Direkthilfe | | Martha M Robbins |
| Africa's Eden | | Martha M Robbins |
| BHP Billiton | | Martha M Robbins |
| Heidelberg Zoo | | Martha M Robbins |
| African Conservation Development Group | | Martha M Robbins |

The funders had no role in study design, data collection and interpretation, or the decision to submit the work for publication. Open access funding provided by Max Planck Society.

## Author contributions

Nikolaos Smit, Conceptualization, Data curation, Formal analysis, Investigation, Visualization, Methodology, Writing – original draft, Writing – review and editing; Martha M Robbins, Data curation, Funding acquisition, Investigation, Methodology, Writing – review and editing

## Author ORCIDs

Nikolaos Smit ⓘ https://orcid.org/0000-0003-0440-1998
Martha M Robbins ⓘ https://orcid.org/0000-0002-6037-7542

## Ethics

We followed the regulations of Agence Nationale des Parcs Nationaux and the Centre National de la Recherche Scientifique et Technique of Gabon as well as the regulations of Uganda Wildlife Authority and the Uganda National Council of Science and Technology in Uganda. Ethical clearance was given by the Max Planck Society.

Joint Public Review: https://doi.org/10.7554/eLife.107093.3.sa1
Author response https://doi.org/10.7554/eLife.107093.3.sa2

# Additional files

## Supplementary files

MDAR checklist

## Data availability

The data and code necessary to replicate this study are available at GitLab, copy archived at *Smit, 2025*.

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
