## [Editor Report · eLife Assessment]

This **important** study uses long-term behavioural observations to understand the factors that influence female-on-female aggression in gorilla social groups. The evidence supporting the claims is **convincing**, as it includes novel methods of assessing aggression and considers other potential factors. The work will be of interest to broad biologists working on the social interactions of animals.

---

## [Referee Report · Joint Public Review]

Summary:

This work aims to improve our understanding of the factors that influence female-on-female aggressive interactions in gorilla social hierarchies, using 25 years of behavioural data from five wild groups of two gorilla species. Researchers analysed aggressive interactions between 31 adult females, using behavioural observations and dominance hierarchies inferred through Elo-rating methods. Aggression intensity (mild, moderate, severe) and direction (measured as the rank difference between aggressor and recipient) were used as key variables. A linear mixed-effects model was applied to evaluate how aggression direction varied with reproductive state (cycling, trimester-specific pregnancy, or lactation) and sex composition of the group. This study highlights the direction of aggressive interactions between females, with most interactions being directed from higher- to lower-ranking adult females close in social rank. However, the results show that 42% of these interactions are directed from lower- to higher-ranking females. Particularly, lactating and pregnant females targeted higher-ranking individuals, which the authors suggest might be due to higher energetic needs, which increase risk-taking in lactating and pregnant females. Sex composition within the group also influenced which individuals were targeted. The authors suggest that male presence buffers female-on-female aggression, allowing females to target higher-ranking females than themselves. In contrast, females targeted lower-ranking females than themselves in groups with a larger ratio of females, which supposes a lower risk for the females since the pool of competitors is larger. The findings provide an important insight into aggression heuristics in primate social systems and the social and individual factors that influence these interactions, providing a deeper understanding of the evolutionary pressures that shape risk-taking, dominance maintenance, and the flexibility of social strategies in group-living species.

The authors achieved their aim by demonstrating that aggression direction in female gorillas is influenced by factors such as reproductive condition and social context, and their results support the broader claim that aggression heuristics are flexible. However, some specific interpretations require further support. Despite this, the study makes a valuable contribution to the field of behavioural ecology by reframing how we think about intra-sexual competition and social rank maintenance in primates.

Strengths:

One of the study's major strengths is the use of an extensive dataset that compiles 25 years of behavioural data and 6871 aggressive interactions between 31 adult females in five social groups, which allows for a robust statistical analysis. This study uses a novel approach to the study of aggression in social groups by including factors such as the direction and intensity of aggressive interactions, which offers a comprehensive understanding of these complex social dynamics. In addition, this study incorporates ecological and physiological factors such as the reproductive state of the females and the sex composition of the group, which allows an integrative perspective on aggression within the broader context of body condition and social environment. The authors successfully integrate their results into broader evolutionary and ecological frameworks, enriching discussions around social hierarchies and risk sensitivity in primates and other animals.

---

## [Author Response]

The following is the authors’ response to the original reviews.

**Reviewer #1 (Recommendations for the authors):**
Suggestions:Although this study has an impressive dataset, I felt that some parts of the discussion would benefit from further explanation, specifically when discussing the differences in female aggression direction between groups with different sex compositions. In the discussion is suggested that males buffer female-on-female aggression and that they 'support' lower-ranking females (see line 212), however, the study only tested the sex composition of the group and does not provide any evidence of this buffering. Thus, I would suggest adding more information on how this buffering or protection from males might manifest (for example, listing male behaviours that might showcase this protection) or referencing other studies that support this claim. Another example of this can be found in lines 223-224, which suggests that females choose lower-ranking individuals when they are presented with a larger pool of competitors; however, in lines 227-228, it's stated that this result contradicts previous work in baboons, which makes the previous claim seem unjustified. I recommend adding other examples from studies that support the results of this paper and adding a line that addresses reasons why these differences between gorillas and baboons might be caused (for example, different social dynamics or ecological constraints). In addition, I suggest the inclusion of physiological data such as direct measures of energy expenditure, caloric intake, or hormone levels, as it would strengthen the claims made in the second paragraph of the discussion. However, I understand this might not be possible due to data or time constraints, so I suggest adding more robust justification on why lactation and pregnancy were used as a proxy for energetic need. In the methods (lines 127-128), it is unclear which phase of the pregnancy or lactation is more energetically demanding. I would also suggest adding a comment on the limitations of using reproductive state to infer energetic need. Lastly, if the data is available, I believe it would be interesting to add body size and age of the females or the size difference between aggressor and target as explanatory variables in the models to test if physiological characteristics influence female-on-female aggression.

Male support:

We have now added more references (Watts 1994, 1997) and enriched our arguments regarding male presence buffering aggression. Previous research suggests that male gorillas may support lower-ranking females and they may intervene in female-female conflicts (Sicotte 2002). Unfortunately, our dataset did not allow us to test for male protection. We conduct proximity scans every 10 minutes and these scans are not associated to each interaction, meaning that we cannot reliably test if proximity to a male influences the likelyhood to receive aggression.

Number of competitors and choice of weaker competitors:

We added a very relevant reference in humans, showing that people choose weaker competitors when they have they can choose. We removed the example to baboons because it used sex ratio and the relevance to our study was not that straightforward.

Reproductive state as a proxy for energetic needs:

We now mention clearly that reproductive state is an indirect measure of energetic needs.

We rephrased our methods to: “Lactation is often considered more energetically demanding than pregnancy as a whole but the latest stages of pregnancy are highly energetically demanding, potentially even more than lactation”

Unfortunately, we do not have access to physiological and body size data. Regarding female age, for many females, ages are estimates with errors up to a decade, and thus, we choose not to use them as a reliable predictor. Having accurate values for all these variables, would indeed be very valuable and improve the predicting power of our study.

Recommendations for writing and presentation:Overall, the manuscript is well-organised and well-written, but there are certain areas that could improve in clarity. In the introduction, I believe that the term 'aggression heuristic' should be introduced earlier and properly defined in order to accommodate a broader audience. The main question and aims of the study are not stated clearly in the last paragraph of the introduction. In the methods, I think it would improve the clarity to add a table for the classification of each type of agonistic interactions instead of naming them in the text. For example, a table that showcase the three intensity categories (severe, mild and moderate), than then dives into each behaviour (e.g. hit, bite, attack, etc.) and a short description of these behaviours, I think this would be helpful since some of the behaviours mentioned can be confusing (what's the difference between attack, hit and fight?). In addition, in line 104, it states that all interactions were assigned equal intensity, which needs to be explained.

We now define aggression heuristics in both the abstract and the first paragraph of the introduction. We have also explained aggressive interactions that their nature was not obvious from their names. Hopefully, these explanations make clear the differences among the recorded behaviours.

We have now specified that the “equal intensity” refers to avoidances and displacements used to infer power relationships: “We assigned to all avoidance/displacement interactions equal intensity, that is, equal influence to the power relationship of the interacting individuals”

Minor corrections:(1) In line 41, there is a 1 after 'similar'. I am unsure if it's a mistake or a reference.

We corrected the typo.

(2) In lines 68-69, there is mention of other studies, but no references are provided.

We added citations as suggested.

(3) Remove the reference to Figure 1 (line 82) from the introduction; the figure should be referenced in the text just before the image, however, your figure is in a different section.

We removed the reference as suggested.

(4) Line 98 and 136, it's written 'ad libtum' but the correct spelling is 'ad libitum'.

We corrected the typo.

(5) Figure 3, remove the underscores between the words in the axis titles.

We removed the underscores.

**Reviewer #2 (Recommendations for the authors):**
Here, I have outlined some specific suggestions that require attention. Addressing these comments will enhance the readability and enhance the quality of the manuscript.(1) L69. Add citation here, indicating the studies focusing on aggression rates.

We added citations as suggested.

(2) L88. The study periods used in this study and the authors' previous study (Reference 11) are different. So please add one table as Table 1 showing the details info on the sampling efforts and data included in their analysis of this study. For example, the study period, the numbers of females and males, sampling hours, the number of avoidance/displacement behaviors used to calculate individual Elo-ratings, and the number of mild/moderate/severe aggressive interactions, etc.

We have now added another table, as suggested (new Table 1) and we have also made clear that we used the hierarchies presented in detail in (Smit & Robbins 2025).

(3) L103. If readers do not look over Reference 25 on purpose, they do not know what the authors want to talk about and why they mention the optimized Elo-rating method. Clarify this statement and add more content explaining the differences between the two methods, or just remove it.

We rephrased the text and in response to the previous comment, we clearly state that there are more details about our approach in Smit & Robbins 2025. At the end of the relevant sentence, we added the following parenthesis “(see “traditional Elo rating method”; we do not use the “optimized Elorating method” as it yields similar results and it is not widely used)” and we removed the sentence referring to the optimized Elo-rating method.

(4) L110. Here, the authors stated that the individual with the standardized Elo-score 1 was the highest-ranking. L117, the "aggression direction" score of each aggressive interaction was the standardized Elo-score of the aggressor, subtracting that of the recipient. So, when the "aggression direction" score was 1, it should mean that the aggressor was the highest-ranking and the recipient was the lowest-ranking female. This is not as the authors stated in L117-120 (where the description was incorrectly reversed). Please clarify.

The highest ranking individual has indeed Elo_score equal to 1 and we calculated the interaction score (or "aggression direction score") of each aggressive interaction by subtracting the standardized Elo-score of the aggressor from that of the recipient (Elo_recepient – Elo_aggressor). So, when the aggressor is the lowest-ranking female (Elo_score=0) and the recipient the highestranking female one (Elo_score=1), the "aggression direction score" is 1-0 = 1.

(5) Regarding point 3 of the Public Review, please also revise/expand the paragraph L193-208 in the Discussion section accordingly.

Please see our response to the public review. We have enriched the results section, added pairwise comparisons in a new table (Table 2) and modified the discussion accordingly.

(6) Table 1. It's not clear why authors added the column 'Aggression Rate' but did not provide any explanation in the Methods/Results section. How did they calculate the correlation between each tested variable and the "overall adult female aggression rates"? Correlating the number of females in the first trimester of female pregnancy with the female aggression rates in each study group? What did the correlation coefficients mean? L202-204 may provide some hints as to why the authors introduced the Aggression Rate. But it should be made clear in the previous text.

We now added more details in the legend of the table to make our point clear: “To highlight that aggression rates can increase due to increase in interactions of different score, we also include the effect of some of the tested variables on overall adult female aggression rates, based on results of linear mixed effects models from (Smit & Robbins 2024).” We did not include detailed methods to calculate those results because they are detailed in (Smit & Robbins 2024). We find it valuable to show the results of both aggression rates and aggression directionality according to the same predictor variables as a means to clarify that aggression rates and aggression directionality are not always coordinated to one another (they do not always change in a consistent manner relative to one another).

(7) L166.This is not rigorous. Please rephrase. There is only one western gorilla group containing only one resident male included in the analysis.

We have toned down our text: “Our results did not show any significant difference between femalefemale aggression patterns within the one western and four mountain gorillas groups”

(8) L167. I don't think the interaction scores in the third trimester of female pregnancy were significantly higher than those in the first trimester. The same concern applies in L194-195.

We have now added a new table with post hoc pairwise comparisons among the different reproductive states that clarifies that.

(9) L202. There is no column 'Aggression rates' in Table 1 of Reference 11.

We have rephrased to make clear that we refer to Table 1 of the present study.

(10) L204-205. Reference 49. Maybe not a proper citation here. This claim requires stronger evidence or further justification. Additionally, please rephrase and clarify the arguments in L204208 for better readability and precision.

We have added three more references and rephrased to clarify our argument.

**Reviewer #3 (Recommendations for the authors):**
(1) Line 41: The word "similar" is misspelled.

We corrected the typo.